# Respective Effects of Oral Hygiene Instructions and Periodontal Nonsurgical Treatment (Debridement) on Clinical Parameters and Patient-Reported Outcome Measures with Respect to Smoking

**DOI:** 10.3390/jcm9082491

**Published:** 2020-08-03

**Authors:** Leila Salhi, Adelin Albert, Laurence Seidel, France Lambert

**Affiliations:** 1Department of Periodontology and Oral Surgery, Faculty of Medicine, University of Liège, 4000 Liège, Belgium; 2Department of Public Health Sciences, University of Liège, 4000 Liège, Belgium; aalbert@uliege.be; 3Department of Biostatistics and Medico-economic information, University of Liège, 4000 Liège, Belgium; laurence.seidel@chuliege.be; 4Dental Biomaterials Research Unit, Department of Periodontology and Oral Surgery, Faculty of Medicine, University of Liège, 4000 Liège, Belgium; France.Lambert@chuliege.be

**Keywords:** periodontitis, oral hygiene, oral health, debridement, periodontal non-surgical treatment

## Abstract

Background: Oral hygiene instructions (OHI) and periodontal nonsurgical treatment (PNST) play pivotal roles in the management of periodontitis. The study aims to discern their respective effects on periodontal clinical parameters and patient-reported outcome measures (PROMs). Methods: Ninety-one patients were included, 34 non-smokers (NS), 25 former smokers (FS) and 32 current smoker (CS). Clinical parameters such as probing depth (PD) and bleeding on probing (BOP) were collected, and the periodontal inflamed tissue area (PISA) was calculated. Clinical parameters and PROMs were recorded before and after receiving OHI, with electronic tooth brush and interdental brushes, as well as 3 months after debridement. Results: Smokers presented a significantly higher proportion of severe periodontitis (64.7%) with generalized extension (76.5%) and with a rapid rate of progression (97.1%) compared to NS and FS. OHI led to a significant decrease of PD, BOP, and PISA *(p* < 0.0001) only in NS and FS. Debridement reduced PD and the percentage of PD >6 mm in all groups (*p* < 0.0001). OHI induced significant improvement of oral hygiene, frequency of interdental cleaning, and PROMs (*p* < 0.0001). Further debridement induced significant additional improvement PROMs in FS and NS *(p* < 0.0001). Conclusion: OHI and debridement improved periodontal clinical parameters and PROMs in both NS and FS. Former smokers had comparable outcomes to non-smokers, suggesting that smoking cessation should be encouraged.

## 1. Introduction

According to the World Health Organization (WHO), oral health contributes to general health and quality of life. Poor oral hygiene and its consequences on the individual such as periodontitis is a recognized public health problem [1,2,3], as periodontal pathogens can reach the bloodstream and contribute to the progression of systemic diseases [4,5,6]. Additionally, in the absence of treatment, periodontal diseases can lead to tooth mobility, tooth loss, and halitosis that negatively affect patient lifestyle with social, physical, and psychological repercussions [7,8,9]. Tobacco use and poor oral hygiene combined an increase the risk of periodontitis not only by initiating but also by fostering disease progression and affecting response to treatment [10,11].

Smoking influences the host inflammatory and immune response and consequently participates in the progression of periodontitis. In a recent systematic review [12], the authors concluded that smoking increases the risk of periodontitis by 85% (risk ratio 1.85, 95% confidence interval: 1.5–2.2). Indeed, the addiction impairs periodontal wound healing, as reviewed by Labriola et al. [13], contributing to disease progression. The effect of initial periodontal therapy in smoking versus non-smoking patients was widely explored in the literature. A better reduction in probing depths, clinical attachment level, bleeding on probing, and plaque index was consistently found in non-smokers compared to smokers [14,15,16,17,18,19,20,21].

Inadequate self-performed plaque control is also recognized as a risk factor of periodontitis [1,22,23], causing the inflammatory destruction of the connective tissue supporting tooth, with gingival bleeding and halitosis, which are considered poor patient-related outcomes [24,25]. In the same context, a recent European Workshop of Periodontology highlighted the role of practitioners in providing oral hygiene instructions and home care in the prevention of periodontal disease [23,26,27].

Beside the systemic management of periodontitis as well as patient information, the initial treatment involves both oral hygiene instructions (OHI) and periodontal nonsurgical treatment (PNST) by a professional mechanical plaque removal including supra and subgingival debridement [2]. However, to the best of our knowledge, the respective effects of these two treatment aspects have never been investigated in the literature. Indeed, most data available concerning the effect of the initial periodontal therapy evaluate the impact of oral hygiene instructions and debridement together. However, discerning the individual impact on oral health of OHI and debridement on clinical parameters and patient-reported outcome measures (PROMs) would help emphasize the respective and synergic roles of the patient and the periodontist.

This controlled prospective study aimed to explore the respective effects of oral hygiene instructions and periodontal debridement on clinical parameters and patient-related outcomes in relation to smoking. 

## 2. Material and Methods

### 2.1. Study Design 

The present study was designed as a single-center, prospective, controlled study focusing on the effect of oral hygiene instructions and PNST on clinical features and PROMs with respect to smoking. The study followed and respected the items of the “STrengthening the Reporting of OBservational studies in Epidemiology (STROBE)” statement for cohort study.

The study was approved on 12 February 2014 by the ethical committee of the University Hospital of Liege, Belgium (B707201421977) and was registered on clinicaltrial.gov (NCT04061460). The goals of the study were carefully explained, and all patients signed an informed consent form (Figure 1: Flowchart).

### 2.2. Study Population

Patients presenting with chronic periodontitis and requiring periodontal treatment were recruited in the Department of Periodontology and Oral Surgery at the University Hospital of Liège, Belgium (CHU, Sart-Tilman) according to the following inclusion criteria: (1) aged at least 18 years old, (2) chronic periodontitis, (3) presence of a minimum of 6 teeth at each arch, (4) a minimum of 6 teeth with pocket depth of 5 mm, and (5) signed informed consent. The exclusion criteria were as follows: (1) aggressive periodontitis, (2) diabetes, (3) connective tissue disease, (4) pregnancy, (5) radiotherapy, (6) chemotherapy, (7) psychological disease, and (8) previous periodontal therapy. Prospective participants were screened for enrolment in the study according to the inclusion and exclusion criteria. Participants who complied with the inclusion criteria were enrolled in the study according to their smoking status: non-smoker (NS), former smoker (FS), or current smoker (CS). They were provided with written information concerning the study requirements.

### 2.3. Clinical Procedure

#### 2.3.1. Oral Hygiene Instructions

After a full initial periodontal examination, all patients received information about the genesis and the overall treatment of periodontitis. Additionally, detailed oral hygiene instructions were provided including teeth brushing, interdental cleaning, and tongue wash according to the gold standard strategies [3,28,29,30]. At the end of the first visit, they received a prescription for an electronic toothbrush and individual interdental brushes. 

#### 2.3.2. Professional Mechanical Plaque Removal

All the patients underwent a professional mechanical plaque removal, involving a supragingival debridement 2 weeks after the first visit and a full mouth subgingival debridement 2 weeks later using ultrasonic piezoelectric device with specific micro-inserts (Newtron P5 XS Bled, Acteon, Satelec, France). Debridement was performed by a single periodontist (L.S).

#### 2.3.3. Smoking Status 

The smoking status was recorded in two ways: (1) the number of cigarettes consumed per day (NCC), and (2) the Fagerström test for nicotine dependence (FTND) [31].

#### 2.3.4. Periodontal Evaluation 

Patients were subjected to a comprehensive periodontal examination including the collection of all periodontal clinical parameters and PROMs at baseline, 2 weeks after OHI, before and after supragingival debridement, and 3 months after the subgingival debridement (Figure 1). 

### 2.4. Data Collection

#### 2.4.1. Periodontal Clinical Measurements

The periodontal examination was conducted by a single periodontist (L.S) and included the following: probing depth (PD), gingival recession (RD), clinical attachment level (CAL), bleeding on probing (BOP), plaque score index (PI), furcation, tooth mobility, percentage of sites of PD ≥6 mm, and the number of missing teeth. A graduated manual periodontal probe (North Carolina 2927.10, Stoma, Germany) was used to take measurements at the 6 sites of each tooth. BOP (%) and plaque score (%) were collected according to Silness and Loe [32]. Tooth mobility was assessed according to the Miller classification with a score ranging from 1 to 3 [33]. The furcation impairments were diagnosed with a Nabers probe according to the classification of Hamp [34], and the periodontal inflamed surface area (PISA) was calculated [35]. Patients were classified according to the new periodontal classification [36] identifying the grade, the stage, and the extent of the periodontitis.

#### 2.4.2. Patient-Reported Outcome Measures (PROMs) 

A questionnaire using a visual analog scale (VAS) was given to all participants regarding their oral hygiene habits and their perception of oral esthetic. It included the following questions scored from 0 to 10: (1) “How do you judge your degree of oral hygiene?” (0: very bad, 10: excellent), (2) “What is the frequency of utilization of interdental brushing?” (0: never, 5: twice per month, 7: once per week, 8: twice per week, 9: third per week, 10: daily), (3) “Do you like the color of your teeth?”(0: not at all, 10: yes, absolutely), (4) “Are your teeth sensitive to cold?”(0: not at all, 10: yes, extremely), (5) “How do you judge your degree of gingival health?”(0: very bad, 10: excellent), (6) “How do you judge the esthetic aspect of your gums?” (0: very low, 10: excellent), (7) “Do you like the color of your gums?” (0: not at all, 10: yes, extremely), (8) “Do your gums bleed during brushing?” (0: not at all, 10: yes, every day), and (9) “How do you judge your breath?” (0: very low, 10: excellent).

All the periodontal clinical parameters, the patient oral hygiene habits, and the patient-centered outcomes were recorded in an online remote controlled and secured database.

### 2.5. Statistical Analyses

The primary outcomes and the null hypothesis of the study were respectively, probing depth (PD) and “no change in the primary endpoint after treatment”. Results were presented as mean and standard deviation (SD) for continuous parameters and as frequency tables for categorical variables. On graphs, mean values were reported with their standard error (SE). Time-related data were analyzed by linear mixed models with time, group, and time × group interactions as fixed factors and patients as random elements. Regression coefficients were given with their standard error (SE). All *p*-values were adjusted by Scheffé’s method for multiple comparisons. Results were considered significant at the 5% critical level (*p* < 0.05). All calculations and graphs were performed with SAS version 9.4 for Windows and R version 3.6.1.

## 3. Results

### 3.1. Patient Characteristics

A total of 91 patients were included in the study (Table 1). Their mean age was 47.3 ± 12.2 years, of which 59.3% were men and 40.7% were women. According to smoking status, patients were distributed as follows: non-smokers (*n* = 32), former smokers (*n* = 25), and smokers (*n* = 34). The three groups did not differ with respect to age and gender. However, smokers presented more frequently with severe (stage IV) periodontitis, potential for dentition loss (62.9%), generalized extension (52.0%), and with a rapid rate of progression (97.1%) compared to the non-smokers and former smokers. 

At baseline (Table 2), smokers also presented with significantly higher PD (*p* = 0.0019), mean and max CAL (*p* < 0.0001), and percentage of PD > 6 mm (*p* = 0.036) compared to the others, and it remained generally so throughout the treatment phases. By contrast, they had lower PISA scores (*p* < 0.001) and BOP% (*p* < 0.0001), but their plaque index was comparable (*p* = 0.20). Periodontitis classification and periodontal clinical parameters were comparable between former smokers and non-smoker patients. PROMs recorded at baseline (Table 3) showed that current smokers had lower interdental scores (*p* = 0.028), esthetic assessment of their gums (*p* < 0.0001) and color of their gums (*p* = 0.0087); otherwise, they were comparable to former smokers and non-smokers. During follow-up, despite oversampling (*n* = 91 instead of *n* = 75), several patients were lost or withdrew from the study, leaving 88 patients (respectively 31, 25, and 32 for current smokers, former smokers, and non-smokers) after OHI (2 weeks) and 50 patients (respectively, 14, 19, and 17) after PNST (3 months). When comparing the baseline (demographic, periodontal, and esthetic) data of completers and lost-to-follow patients in each group, no significant differences were found in their characteristics.

### 3.2. Effects on Periodontal Clinical Parameters 

As seen in Table 2 and Figure 2, oral hygiene instructions led to a highly significant decrease of PISA and mean BOP (%) values (*p* < 0.0001) in former and non-smokers but in not in current smokers (*p* = 0.94). OHI also significantly reduced the mean plaque index in all three groups (*p* < 0.0001) with reductions of 74.8% in former smokers, 64.7% in non-smokers, and 45.4% in smokers. The other periodontal parameters (mean and max PD, mean and max Cal, % PD > 6 mm) remained unchanged. Full mouth subgingival debridement induced an additional significant reduction (*p* < 0.0001) in all periodontal parameters in all three smoking groups except for PISA, Mean BOP (%), and mean plaque index where values remained unchanged. Of important note, the mean PD dropped by 1.44 mm in smokers, 1.30 mm in former smokers, and by 1.07 mm in non-smokers. Similarly, max PD decreased by 1.68, 3.08, and 2.96 mm, respectively. The percentage of PD >6 mm went down from 18.2 to 2.92 in current smokers, 12.7 to 0.25 in former smokers, and 11.7 to 0.42 in non-smokers.

### 3.3. Effects on PROMs 

The respective effects OHI and debridement on PROMs are described in Table 3 and Figure 3. OHI positively and markedly influenced patient reporting of their oral hygiene degree and of the frequency of interdental cleaning in the 3 groups (all adjusted *p*-values < 0.0001). The esthetic aspect of the gum was also significantly improved in former and non-smokers (*p* < 0.0001) but not in current smokers (*p* = 0.54). All the other items remained unchanged in each group. According to patient reporting, debridement did not further improve PROMs in smokers, except for a borderline effect on teeth color (*p* = 0.048). By contrast, debridement significantly enhanced teeth color, sensitivity, gingival health, bleeding on brushing, and good breath/halitosis scores in former smokers and non-smokers (all *p* < 0.0001). 

## 4. Discussion

To the best of our knowledge, the present study is the first to discern the separate effects of oral hygiene instructions and debridement on periodontal clinical parameters and PROMs according to smoking status (non-smokers, former smokers, and current smokers). 

On the one hand, the oral hygiene instructions were responsible for a significant decrease (>50%) of the mean plaque index in the three groups of PISA and BOP (%) only in non-smokers (NS) and former smokers (FS). Indeed, higher reductions of plaque index were found in these two groups compared to the smoker group. In the case of gingivitis, it was widely demonstrated that plaque control significantly improves gingival inflammation and BOP [3]. In addition, a recent systematic review confirmed that the mechanical plaque control procedures are effective in reducing plaque and gingivitis [22]. Moreover, in non-smokers and former smokers, the oral hygiene improvement contributed to the control of the PI, which participates in the decrease of BOP and hence of the PISA [3]. As known, the interactions among human oral bacteria, called the oral microbiome, are integral to the development and maturation of the plaque [37]. Therefore, the NS and FS groups with higher compliance to plaque control presented a higher control of their oral microbiome than the CS group. However, the study highlighted a lower compliance of smokers with respect to instructions on dental hygiene. The baseline BOP and PISA values of smokers were already low and were not influenced by the improvement of plaque control. This observation can be explained by the fact that smoking suppresses the inflammatory gingival response to the plaque accumulation as well as the gingival bleeding (BOP) [38]. Furthermore, in a study investigating host response during experimental gingivitis, the authors did also observe that smokers had less bleeding and a higher proportion of visible plaque index (VPI) than non-smokers [39]. Of note, OHI had no influence whatsoever on PD and CAL.

On the other hand, the debridement led to a significant reduction of the mean PD (>1 mm), which is the primary outcome measure of this study, and of the percentage of PD >6 mm (>10%) in all three groups (*p* < 0.0001), also with a higher reduction in non-smokers and former smokers. The same observation was also made for the clinical attachment level. The reduction of PI, BOP, and PD after initial periodontal therapy was widely demonstrated in clinical studies [15,16,17,18,19,20,21] and systematic reviews [40,41]. However, the present study highlights the effectiveness of plaque control (PI) on both the inflammation level (BOP) and probing depth reduction in non-smokers and former smokers, while in smokers, periodontal debridement solely improves clinical periodontal parameters. These findings also emphasized the benefits of smoking cessation on periodontal outcomes as recently described in a WHO systematic review and meta-analysis on the effects of tobacco use cessation [42].

Clinical outcomes after both OHI and PNST in smokers were limited, which could be attributed to the impact of smoking on inflammatory and immune host responses and also to the negative vasoconstrictive effect of nicotine. This addiction decreases the phagocytes population [43,44], the number and the chimiotactisms of neutrophils [45,46], and it interacts with the lymphocytes blood circulation [47,48] with the consequence of low bacteria destruction. This could explain the lower reduction of PI in smokers. Moreover, since the vasoconstrictive effect of tobacco smoke [49] leads to the diminution of blood cells in the capillaries, no significant reduction of BOP could be observed in smokers, neither after OHI nor after PNST. Additionally, nicotine induces the apoptosis of fibroblasts [50,51] that also perturbs wound healing and the pocket depth reduction after debridment. It is relevant to note that the present controlled prospective study highlighted the positive effect of smoking cessation to the response of initial periodontal therapy. Our findings confirm those of a recent systematic review which concluded that in former smokers, the risk for periodontitis becomes comparable to never-smokers and that nonsurgical periodontal treatment improves clinical outcomes after smoking cessation [12].

Patient-centered outcome assessment is now being considered as a primary outcome measure in clinical trials, as described in a recent systematic review [52]. In the extant literature on PROMS after PNST, the authors mostly use a visual analog scale (VAS) and questionnaires to evaluate patient’s quality of life [53,54], level of anxiety, and/or the pain [54,55,56]. However, there is a clear lack of information concerning the patient’s esthetic perception of gums, bleeding [57], and color of gums and teeth, which are related simultaneously to OHI. In the present study, all patients were compliant to OHI, and most PROMs scores improved except for the patient’s perception of his/her “gum esthetic”, which significantly decreased in former smokers and non-smokers. This could be explained by the fact that at baseline, the initial questionnaire was filled in before any periodontal clinical measurements or any information about periodontitis genesis and classification was available; therefore, at the second visit, when patients became aware of their gum situation, they tended to express a worst perception. The present study clearly evidenced that after debridement, outcome measures such as teeth color, sensitivity, gingival health, bleeding on brushing, and good breath/halitosis were considerably improved in the patient’s perception but only in former smokers and non-smokers. Perception did not change in current smokers. Of note, the patient’s degree of interdental hygiene tends to decline with time with even less compliance in smokers, so that it is essential to promote the necessity of maintenance recall. Specifically, patients should be compliant in their oral health and in the follow-up of their periodontal treatment, and practitioners should employ the motivational counseling and microbiome control through supra debridement. Decades ago, some authors [58] stressed the role of the maintenance care program to prevent the recurrence of periodontitis. Some years later, the authors made a long-term study of 30 years and concluded that preventive dental treatment was beneficial for maintaining a high standard of oral hygiene. Moreover, for the compliant patient, the incidence of caries and periodontal disease as well as tooth mortality was quite low [59]. In a systematic review of 19 studies evaluating the efficacy of long-term professional mechanical plaque removal [60], the authors concluded that the periodontal maintenance called “supportive therapy” may limit the incidence and yearly rate of tooth loss as well as the loss in clinical attachment in patients treated for periodontitis. Additionally in a workshop on primary and secondary prevention of periodontal and peri-implant diseases [2], the authors concluded that professional mechanical plaque removal as the sole element of professional preventive care is inappropriate without education and behavioral changes to sustained improvements in health status.

The main limitation of the present study could be related to this proportion of drop-outs after the periodontal subgingival debridement, despite a larger sample size (*n* = 91) than foreseen (*n* = 75) at enrollment to cope with the potential loss of subjects. This shortcoming may be attenuated by the following arguments: (1) all patients (completers and drop-outs) received the entire treatment sequence (OHI + PNST), (2) no significant differences were found between completers and drop-outs at baseline, (3) the statistical analysis used linear mixed effects models which included all longitudinal data available of complete and drop-outs subjects, (4) all *p*-values from the linear mixed models were adjusted for multiple comparisons, and (5) despite drop-outs, the null hypothesis of no change in the primary endpoint “probing depth (PD)” after treatment was rejected at a high significance level (adjusted *p*-value < 0.0001) with a mean observed effect size >1 mm in each group.

## 5. Conclusions

Oral hygiene instructions and PNST combined play pivotal roles in the global periodontal treatment. Separate improvements of periodontal clinical outcomes and PROMs due respectively to OHI and PNST were observed in non-smokers and former smokers. In current smokers, improvements were only visible after PNST. Periodontists should welcome and encourage smoking cessation by supporting the idea that former smokers have comparable outcomes to non-smokers. Moreover, there is a need to promote the impact of recall maintenance in the promotion of good oral health. Finally, long-term multicentric studies will be needed to evaluate the contribution of OHI in the success of periodontal non-surgical treatment and to assess quality of life.

## Figures and Tables

**Figure 1 jcm-09-02491-f001:**
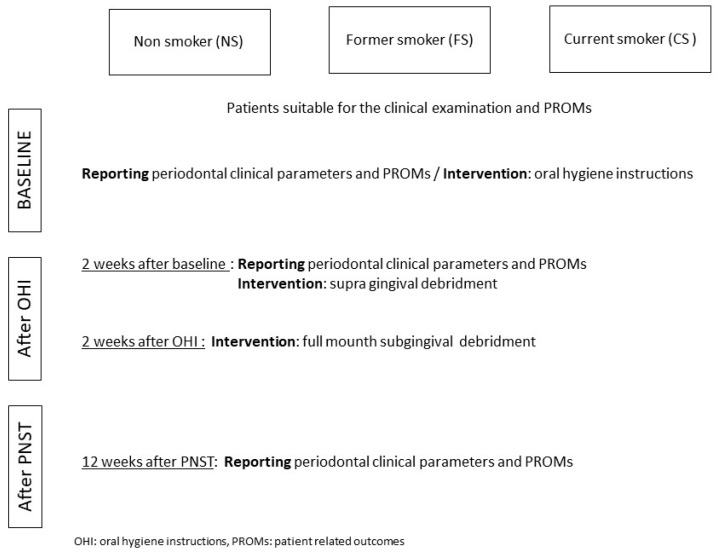
Flowchart: study design.

**Figure 2 jcm-09-02491-f002:**
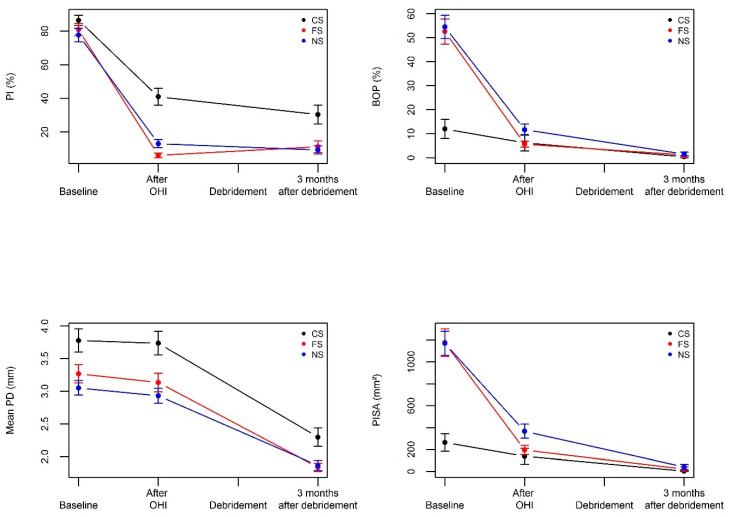
Effect of oral hygiene instructions (OHI) and debridement on periodontal clinical outcomes in current smokers, former smokers, and non-smokers.

**Figure 3 jcm-09-02491-f003:**
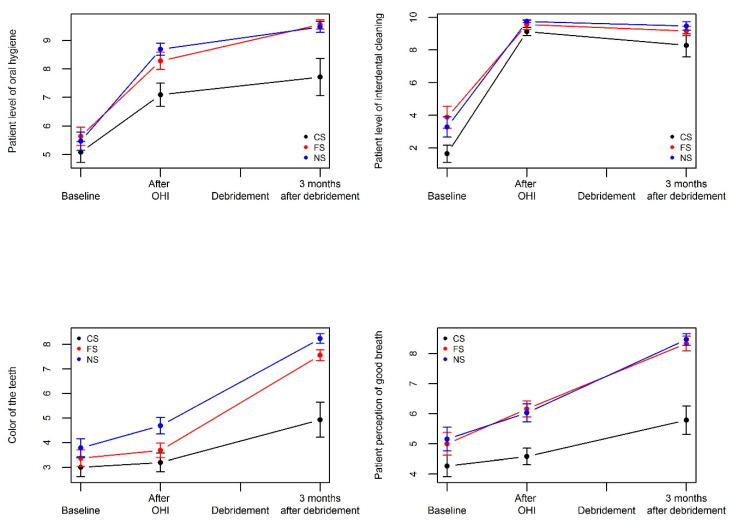
Effect of OHI and debridement on PROMs in current smokers, former smokers and non-smokers.

**Table 1 jcm-09-02491-t001:** Patient demographic characteristics, periodontitis classification, and smoking status

Variable	Category	Total	Current Smoker	Former Smoker	Non-Smoker	*p*-Value
		*n* = 91	*n* = 34	*n* = 25	*n* = 32	
Demography						
Age (mean ± SD, years)		47.3 ± 12.2	46.5 ± 11.5	51.2 ± 11.6	45.0 ± 12.9	0.14
Gender (*n*, %)						0.84
	Female	54 (59.3)	20 (58.8)	16 (64.0)	18 (56.3)	
	Man	37 (40.7)	14 (41.2)	9 (36.0)	14 (43.8)	
Dental characteristics						
Missing teeth (*n*, %)		446 (15.4)	230 (21.3)	112 (14.1)	104 (10.3)	<0.0001
Periodontitis classification						
Stage (*n*; %)						
	III	60 (65.9)	15 (44.1)	19 (76.0)	26 (81.3)	0.0029
	IV	31 (34.1)	19 (55.9)	6 (24.0)	6 (18.7)	
Extent (*n*, %)						
	L	42 (46.2)	9 (26.5)	14 (56.0)	19 (59.4)	0.011
	G	48 (52.8)	25 (73.5)	11 (44.0)	12 (37.5)	
	M-I	1 (1.1)	0 (0.0)	0 (0.0)	1 (3.1)	
Grade (*n*, %)						
	A	44 (48.3)	0 (0.0)	18 (72.0)	26 (81.2)	<0.0001
	B	15 (16.5)	2 (5.9)	7 (28.0)	6 (18.8)	
	C	32 (35.2)	32 (94.1)	0 (0.0)	0 (0.0)	

**Table 2 jcm-09-02491-t002:** Effect of oral hygiene instructions and debridement on periodontal clinical parameters

Characteristic	Baseline	After Oral Hygiene Instructions ^(b)^	After Debridement ^(c)^
Population			
Current smoker	34	31	14
Former smoker	25	25	19
Non-smoker	32	32	17
Variable ^(a)^ (mean ± SD)			
PISA (mm²)			
CS	265 ± 470	139 ± 404	4.22 ± 15.8
FS	1176 ± 635	195 ± 207 *	22.8 ± 88.7
NS	1170 ± 632	366 ± 364 *	40.9 ± 94.3
	(*p* < 0.001) ^(d)^	(*p* = 0.028)	(*p* = 0.44)
Mean PD (mm)			
CS	3.78 ± 1.05	3.74± 1.01	2.30 ± 0.53 *
FS	3.27 ± 0.70	3.14 ± 0.72	1.84 ± 0.28 *
NS	3.05 ± 0.63	2.93 ± 0.66	1.86 ± 0.33 *
	(*p* = 0.0019)	(*p* = 0.0006)	(*p* = 0.0021)
Max PD (mm)			
CS	8.00 ± 1.21	7.97 ± 1.22	6.29 ± 1.90 *
FS	7.80 ± 1.19	7.76 ± 1.16	4.68 ± 1.16 *
NS	7.72 ± 1.11	7.72 ± 1.11	4.76 ± 0.97 *
	(*p* = 0.61)	(*p* = 0.67)	(*p* = 0.0027)
Variable * (mean ± SD)			
Mean CAL (mm)			
CS	4.34 ± 1.40	4.17 ± 1.34	2.72 ± 0.67 *
FS	3.40 ± 0.81	3.26 ± 0.84	1.88 ± 0.30 *
NS	3.10 ± 0.68	2.97 ± 0.69	1.92 ± 0.38 *
	(*p* < 0.0001)	(*p* < 0.0001)	(*p* < 0.0001)
Max CAL (mm)			
CS	9.56 ± 2.48	9.26 ± 2.38	8.14 ± 2.71 *
FS	8.16 ± 1.52	8.12 ± 1.51	5.11 ± 1.20 *
NS	8.06 ± 1.46	7.91 ± 1.30	5.00 ± 1.22 *
	(*p* = 0.0032)	(*p* = 0.0091)	(*p* < 0.0001)
Mean BOP (%)			
CS	12.0 ± 23.0	6.24 ± 18.5	0.42 ± 1.58
FS	52.6 ± 26.1	5.66 ± 6.08 *	1.21 ± 4.69
NS	54.5 ± 27.3	11.7 ± 13.3 *	1.55 ± 3.22
	(*p* < 0.0001)	(*p* = 0.19)	(*p* = 0.67)
Mean PI (%)			
CS	86.4 ± 17.7	41.0 ± 28.2 *	30.4 ± 21.1
FS	80.9 ±17.8	6.11 ± 7.05 *	11.3 ± 15.3
NS	77.8 ± 22.9	13.1 ± 13.0 *	9.38 ± 10.1
	(*p* = 0.20)	(*p* < 0.0001)	(*p* = 0.0009)
% PD > 6 mm			
CS	19.0 ± 16.0	18.2 ± 15.5	2.92 ± 3.45 *
FS	12.9 ± 8.71	12.7 ± 8.54	0.25 ± 0.75 *
NS	11.9 ± 7.61	11.7 ± 7.30	0.42 ± 1.19 *
	(*p* = 0.036)	(*p* = 0.054)	(*p* = 0.0007)

^(a)^ CS = current smoker, FS = former smoker, NS = non-smoker, PISA = periodontal inflamed surface area, PD = probing depth, CAL = clinical attachment level, BOP = bleeding on probing, PI = plaque index; percent, mean and max values derived from 6 sites measurements for each tooth. ^(b)^ * the asterisk indicates a significant change from baseline after oral hygiene instructions (all *p*-values derived by linear mixed models were less than 0.0001 adjusted for multiple comparisons). ^(c)^ * the asterisk indicates a significant change from after debridement compared to after oral hygiene instructions (all *p*-values derived by linear mixed models were less than 0.0001 adjusted for multiple comparisons). ^(d)^ The *p*-values in parentheses assess the comparison of groups and at time point (baseline, after oral hygiene instructions, and after debridement).

**Table 3 jcm-09-02491-t003:** Effect of oral hygiene instructions and debridement on patient-reported outcome measures (PROMs)

PROMs (Mean ± SD) ^(a)^	Group	Baseline	After Oral Hygiene Instructions ^(b)^	After Debridement ^(c)^
Degree of oral hygiene	CS	5.1 ± 2.1	7.1 ± 2.2 *	7.7 ± 2.5
	FS	5.6 ± 1.6	8.3 ± 1.5 *	9.6 ± 0.70
	NS	5.5 ± 1.8	8.7 ± 1.2 *	9.5 ± 0.80
		(*p* = 0.51) ^(d)^	(*p* = 0.0013)	(*p* = 0.0013)
Interdental hygiene	CS	1.6 ± 3.1	9.1 ± 1.4 *	8.3 ± 2.7
	FS	3.9 ± 3.4	9.6 ± 1.5 *	9.2 ± 1.2
	NS	3.3 ± 3.5	9.8 ± 0.4 *	9.5 ± 1.1
		(*p* = 0.028)	(*p* = 0.11)	(*p* = 0.16)
Teeth color	CS	3.0 ± 2.2	3.2 ± 2.1	4.9 ± 2.7 *
	FS	3.4 ± 1.7	3.7 ± 1.5	7.6 ± 0.92 *
	NS	3.8 ± 2.1	4.7 ± 1.9	8.2 ± 0.83 *
		(*p* = 0.31)	(*p* = 0.0075)	(*p* < 0.0001)
Sensitivity	CS	3.2 ± 2.6	3.6 ± 2.2	5.0 ± 2.3
	FS	3.6 ± 1.6	3.8 ± 1.5	7.3 ± 1.0 *
	NS	4.0 ± 1.9	4.6 ± 1.7	7.8 ± 1.2 *
		(*p* = 0.29)	(*p* = 0.076)	(*p* < 0.0001)
Gingival health	CS	4.3 ± 2.5	4.2 ± 2.3	5.1 ± 1.8
	FS	4.2 ± 1.7	4.2 ± 1.8	7.4 ± 1.0 *
	NS	5.0 ± 1.6	5.2 ± 1.5	7.9 ± 1.0 *
		(*p* = 0.20)	(*p* = 0.058)	(*p* < 0.0001)
Esthetic assessment of the gums	CS	3.1 ± 3.0	2.0 ± 1.9	0.86 ± 1.3
	FS	6.0 ± 3.1	1.2 ± 0.88 *	0.61 ± 1.9
	NS	7.3 ± 2.5	1.0 ± 0.76 *	0.18 ± 0.73
		(*p* < 0.0001)	(*p* = 0.0086)	(*p* = 0.39)
Color of the gums	CS	2.8 ± 2.2	3.0 ± 2.0	3.9 ± 2.5
	FS	4.1 ± 2.1	4.4 ± 2.1	5.2 ± 2.2
	NS	4.6 ± 2.5	5.0 ± 2.3	5.8 ± 2.0
		(*p* = 0.0087)	(*p* = 0.0011)	(*p* = 0.068)
Bleeding on brushing	CS	4.6 ± 2.9	4.6 ± 3.0	3.5 ± 3.1
	FS	5.2 ± 2.0	4.8 ± 2.3	0.89 ± 1.5 *
	NS	3.7 ± 2.6	3.3 ± 2.6	0.41 ± 1.2 *
		(*p* = 0.088)	(*p* = 0.049)	(*p* = 0.0002)
Good breath/Halitosis	CS	4.3 ± 2.0	4.6 ± 1.5	5.8 ± 1.8
	FS	5.0 ± 1.9	6.2 ± 1.3	8.3 ± 1.0 *
	NS	5.2 ± 2.2	6.0 ± 1.7	8.5 ± 0.80 *
		(*p* = 0.19)	(*p* = 0.0002)	(*p* < 0.0001)

^(a)^ CS = current smoker, FS = former smoker, NS = non-smoker; Questionnaire: “1/How do you judge your degree of oral hygiene?” (0: very bad, 10: excellent), 2/ “What is the frequency of utilization of interdental brushing?” (0: Never, 5: twice per month, 7: once per week, 8: twice per week, 9: three times per week, 10: daily), 3/ “Do you like the color of your teeth?” (0: Not at all, 10: Yes, absolutely), 4/“Are your teeth sensitive to cold?”(0: Not at all, 10: Yes, extremely), 5/ “How do you judge your degree of gingival health?”(0: very bad, 10: excellent), 6/ “How do you judge the esthetic aspect of you gums?” (0: very low, 10: excellent), 7/ “Do you like the color of your gums?” (0: Not at all, 10: Yes, extremely), 8/ “Do your gums bleed during brushing?” (0: Not at all, 10: Yes, every day), 9/ “How do you judge your breath?” (0: very low, 10: excellent). ^(b)^ * the asterisk indicates a significant change from baseline after oral hygiene instructions (all *p*-values derived by linear mixed models were less than 0.0001 adjusted for multiple comparisons). ^(c)^ * the asterisk indicates a significant change from after debridement compared to after oral hygiene instructions (all *p*-values derived by linear mixed models were less than 0.0001 adjusted for multiple comparisons). ^(d)^ The *p*-values in parentheses assess the comparison of groups and at each time point (baseline, after oral hygiene instructions, and after debridement).

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
