# Peer review of "Respective Effects of Oral Hygiene Instructions and Periodontal Nonsurgical Treatment (Debridement) on Clinical Parameters and Patient-Reported Outcome Measures with Respect to Smoking"

_jcm, 2020, doi:10.3390/jcm9082491_

Round 1

Reviewer 1 Report

Well-designed controlled study on the effect of oral hygiene instructions and PNST on clinical features and PROMs with respect to smoking.

From a methodological point of view, a weak aspect of the research is the possible lower compliance of smokers with the instructions on dental hygiene. This aspect needs a comment in the "Discussion" section.

Another point to be properly developed, in the discussion section, is the reference to the microbiome without any previous comment or reference.

line 195 trivial error: in former and non-smokers but in not in current smokers

Author Response

I thank the reviewers for their critical review and constructive comments.

We carefully revised our manuscript entitled “Respective effects of oral hygiene instructions and periodontal non-surgical treatment (debridement) on clinical parameters and patient-reported outcome measures with respect to smoking” (jcm-872386), according to the reviewers’ suggestions and hope it is now acceptable for publication.

Looking forward to hearing from you,

Kind regards,

Leila Salhi

REVIEWER 1

  1. Well-designed controlled study on the effect of oral hygiene instructions and PNST on clinical features and PROMs with respect to smoking.

Dear reviewer, thank you for your comment.

  1. From a methodological point of view, a weak aspect of the research is the possible lower compliance of smokers with the instructions on dental hygiene. This aspect needs a comment in the "Discussion" section.

Thank you for this relevant concern. A comment has been added in the “Discussion” section as follows (289-291). ”Of note, the patient’s degree of interdental hygiene tends to decline with time with even less compliance in smokers, so that it is essential to promote the necessity of maintenance recall.”

  1. Another point to be properly developed, in the discussion section, is the reference to the microbiome without any previous comment or reference.

To answer this relevant comment we added the following on oral microbiome and plaque index in the text (239-244). “As known, the interactions among human oral bacteria, called the oral microbiome, are integral to the development and maturation of the plaque37. Therefore, the NS and FS groups with higher compliance to plaque control presented a higher control of their oral microbiome than the CS group. However, the study highlighted a lower compliance of smokers with respect to instructions on dental hygiene”

  1. line 195 trivial error: in former and non-smokers but in not in current smokers

Correct!. (204)

Reviewer 2 Report

This is a well presented paper on an important topic. 

Abstract

  • Please add the number of participants.
  • Also the methods need clarification around OHI as the particpants were also provided with a free electric tooth brush (this could be causing any instead of the OHI). 

Introduction

 Two reference suggestions for the introduction: 

  1. WHO monograph on smoking and oral health- https://www.who.int/tobacco/publications/smoking_cessation/monograph-tb-cessation-oral-health/en/
  2. Chambrone 2013 - https://pubmed.ncbi.nlm.nih.gov/23590649/ 

Methods 

  • How did you manage with those using e-cigarettes? which category did they fall into? 
  • Was there any biochemical validation of smoking status. If not please detail that this was self-reported. 
  • No mention of primary outcome measure.
  • No mention of null hypothesis.
  • Sample size caluation needs more information to be understood. The MCID of 0.5mm difference in probing depth is stated but no justification or reference is provided for why this was chosen. What is the variability of the outcome measure (SD)? What outcome does the sample size relate to? OHI? RSI? You mention a primary outcome measure and null hypothesis in the discussion section for the first time. I'd suggest you have slightly confused methods here. I'd recommend removing the sample size calculation and describing this as an exploratory study (which matches what you have done) as I suspect you don't have the information to conduct a proper sample size. 

Discussion

  • You didn't use a oral health quality of life measure? It would be useful to explore this in the discussion. This would be a useful addition to future studies. 
  • You mention several times the vasoconstrictive effect of nicotine. My understanding of the situiation is that it's not as clear as this. Tobacco smoke certainly has significant impairment of the periodontal vasculature, although the specific role of nicotine in this is unclear. I'd recommend a broader term here rather than vasoconstriction. 
  • In the limitations section you talk about an 'expected difference' whereas in the sample size you talk about a MCID. These are different. Again I'd suggest removing the sample size calaculation and this discussion of it in the limitations. 

Well done on a nice study and I hope my comments are useful. 

Author Response

I thank the reviewers for their critical review and constructive comments.

We carefully revised our manuscript entitled “Respective effects of oral hygiene instructions and periodontal non-surgical treatment (debridement) on clinical parameters and patient-reported outcome measures with respect to smoking” (jcm-872386), according to the reviewers’ suggestions and hope it is now acceptable for publication.

Looking forward to hearing from you,

Kind regards,

Leila Salhi

REVIEWER 2

  1. This is a well presented paper on an important topic.

Dear reviewer, thank you for the positive comment.

  1. Abstract
  2. Please add the number of participants.

The number of patients has added as following (17-18): “Ninety-one patients were included, 34 non-smokers (NS), 25 former smokers (FS) and 32 current smokers (CS)”

  1. Also the methods need clarification around OHI as the participants were also provided with a free electric tooth brush (this could be causing any instead of the OHI).

Thank you for the relevant comment. The abstract has modified as follows (19-21). Clinical parameters and PROMs were recorded before and after receiving OHI, with electronic tooth brush and interdental brushes, as well as 3 months after debridement.”

  1. Introduction

  1. Two reference suggestions for the introduction:

WHO monograph on smoking and oral health- https://www.who.int/tobacco/publications/smoking_cessation/monograph-tb-cessation-oral-health/en/

Chambrone 2013 - https://pubmed.ncbi.nlm.nih.gov/23590649/

Dear reviewer, thank you for sharing this reference. We have included it in the paper and made the following addition in the “Discussion” (258-260) . “These findings also emphasized the benefits of smoking cessation on periodontal outcomes as recently described in a WHO systematic review and meta-analysis on the effects of tobacco use cessation42.”

Since the systematic review of Chambrone et al. concerned only two randomized controlled trial we chose not include it in our article.

  1. Methods

  1. How did you manage with those using e-cigarettes? Which category did they fall into?

Dear reviewer, we decided not to include e-cigarettes smokers in the study. Indeed, it is difficult to discriminate the different types of e-cigarette (different concentration of nicotine, without nicotine, and/or additive products). However, in future studies, it may indeed be interesting to analyze all types of smoking consumption.

  1. Was there any biochemical validation of smoking status. If not please detail that this was self-reported.

Thank you for the comment. We added the following in the Material and Methods section (104-106):

2.3.3. Smoking status

The smoking status was recorded in two ways: (1) the number of cigarette consumed per day (NCC), and (2) the Fagerström test for nicotine dependence (FTND)1

  1. No mention of primary outcome measure.
  2. No mention of null hypothesis.

This is correct. The following has been added to the text (138-139): ”The primary outcome was defined as the probing depth (PD) and the null hypothesis stated “No change in PD after treatment”.

  1. Sample size calculation needs more information to be understood. The MCID of 0.5mm difference in probing depth is stated but no justification or reference is provided for why this was chosen. What is the variability of the outcome measure (SD)? What outcome does the sample size relate to? OHI? RSI? You mention a primary outcome measure and null hypothesis in the discussion section for the first time. I'd suggest you have slightly confused methods here. I'd recommend removing the sample size calculation and describing this as an exploratory study (which matches what you have done) as I suspect you don't have the information to conduct a proper sample size.

We decided to follow the reviewer’s recommendation to remove the sample size (power) calculation from the statistical section (139-142).

  1. Discussion

  1. You didn't use a oral health quality of life measure? It would be useful to explore this in the discussion. This would be a useful addition to future studies.

Thank you for the relevant comment and advice for further prospective studies. We made the following addition in the “Conclusion” section (324-325): “Finally, long-term multicentric studies will be needed to evaluate the contribution of OHI in the success of periodontal non-surgical treatment and to assess quality of life”

  1. You mention several times the vasoconstrictive effect of nicotine. My understanding of the situation is that it's not as clear as this. Tobacco smoke certainly has significant impairment of the periodontal vasculature, although the specific role of nicotine in this is unclear. I'd recommend a broader term here rather than vasoconstriction.

Following your point, we suggest to modify the text as follows (266-268): “Moreover, since the vasoconstrictive effect of tobacco smoke leads to the diminution of blood cells in the capillaries, no significant reduction of BOP could be observed in smokers, neither after OHI nor after PNST”

  1. In the limitations section you talk about an 'expected difference' whereas in the sample size you talk about a MCID. These are different. Again I'd suggest removing the sample size calculation and this discussion of it in the limitations.

In agreement with Point (e) above, we slide change argument (4) in the limitation section and discarded any reference to power calculation. Thus, lines 313-316 were changed as follows:

4) despite drop-outs, the null hypothesis of no change in the primary endpoint “probing depth (PD)” after treatment was rejected at a high significance level (adjusted p-value <0.0001) with a mean observed effect size >1 mm in each group.

  1. Well done on a nice study and I hope my comments are useful.

Thank so much dear Reviewer; your precious comments did clearly improve the paper considerably.

  1. Salhi L. Fagerström test for nicotine dependence as an indicator for tobacco-related studies, accepted in the Journal of periodontology. 2020.
  2. Mavropoulos A, Aars H and Brodin P. Hyperaemic response to cigarette smoking in healthy gingiva. J Clin Periodontol. 2003;30:214-21.